# On the Development of a Wearable Animal Monitor

**DOI:** 10.3390/ani13010120

**Published:** 2022-12-28

**Authors:** Luís Fonseca, Daniel Corujo, William Xavier, Pedro Gonçalves

**Affiliations:** 1Departamento de Eletrónica Telecomunicações e Informática and Instituto de Telecomunicações, Universidade de Aveiro, 3830-193 Aveiro, Portugal; 2iFarmTec—Intelligent Farm Technologies, 3830-527 Gafanha da Encarnação, Portugal; 3Escola Superior de Tecnologia e Gestão de Águeda and Instituto de Telecomunicações, Universidade de Aveiro, 3830-193 Aveiro, Portugal

**Keywords:** animal monitoring, wearable sensors, machine learning, behavior prediction

## Abstract

**Simple Summary:**

The choice of sensors for detecting animal behavior through wearable devices has the greatest impact on the quality of the behavior learning process and the cost of production. The present study evaluated, through a machine learning process, the most important features in the detection and classification of sheep behavior, as well as the individual contribution of each of the sensors included in the monitoring collar. The study showed that the gyroscope had a very low contribution towards improving accuracy, despite its high production cost, and also shares very similar characteristics with the accelerometer used in the collar. Conversely, the thermometer, which was intended for other monitoring scenarios, proved to be essential in detecting some states, especially the ones related to postures in which the animal tends to wrap the collar and thus increases the temperature of the device. The final feature set provided by the thermometer and accelerometer was considered a good basis for building an animal behavior monitoring system.

**Abstract:**

Animal monitoring is a task traditionally performed by pastoralists, as a way of ensuring the safety and well-being of animals; a tremendously arduous and lonely task, it requires long walks and extended periods of contact with the animals. The Internet of Things and the possibility of applying sensors to different kinds of devices, in particular the use of wearable sensors, has proven not only to be less invasive to the animals, but also to have a low cost and to be quite efficient. The present work analyses the most impactful monitored features in the behavior learning process and their learning results. It especially addresses the impact of a gyroscope, which heavily influences the cost of the collar. Based on the chosen set of sensors, a learning model is subsequently established, and the learning outcomes are analyzed. Finally, the animal behavior prediction capability of the learning model (which was based on the sensed data of adult animals) is additionally subjected and evaluated in a scenario featuring younger animals. Results suggest that not only is it possible to accurately classify these behaviors (with a balanced accuracy around 91%), but that removing the gyroscope can be advantageous. Results additionally show a positive contribution of the thermometer in behavior identification but evidences the need for further confirmation in future work, considering different seasons of different years and scenarios including more diverse animals’ behavior.

## 1. Introduction

Animal monitoring as a way of studying animal behavior has been receiving a huge effort from academia and industry [1,2]. The objective is to study production processes and conditions not only to improve welfare, but also to increase gains by optimizing operational processes. Different kinds of sensor technologies, and the consequent transport of sensed data for analysis and forecasting in line with the trend of the Internet of Things (IoT), are being applied in the agricultural sector. In the case of animals, one of the technologies that has been greatly developed is related to wearable sensors, due to their low cost, reduced impact on animal lives and because they allow animal behavior to be inferred. For example, by using inertial sensors [2], it is possible to measure animal activity and identify a set of behaviors such as eating, ruminating, walking or resting. From there, it is possible to infer a large amount of information about animals, such as prostration due to illness, heat, or even to measure resting periods. The information collected by this type of sensors is typically transferred to computational infrastructures equipped with Machine Learning (ML) algorithms, either in real-time or through opportunistic communication means, with the data being analyzed and compared and predictions being made about its future evolution.

However, forecasting and analysis depend on the ability to capture relevant information regarding animal behavior, with such ability depending on the quality of the sensors and even the way they are applied to animals. Inertial sensors [3,4], such as accelerometers, magnetometers, or gyroscopes, can sense different physical quantities and are available in a wide range of technologies, qualities, and prices. Regarding the latter, there is a huge disparity in prices of inertial sensors, with values ranging from less than $1.5 for accelerometers to nearly $7 for gyroscopes. Such details are of the utmost importance for the development of a monitoring collar, making sensor selection decisions a complex task. Under this setting, can it be claimed that the higher cost of the gyroscope is compensated for by improving learning outcomes in a computational learning process?

The present work is part of the development of an animal monitoring collar and reports on the evaluation of different inertial sensors’ ability to predict animal behavior. The work includes an individual evaluation of each of the sensors in the accuracy of the learning process, as well as in the complexity of the behavior classification algorithm, thus allowing to base the choice of sensors on a cost-benefit analysis.

The remainder of this paper is organized as follows. Section 2 overviews the related work regarding animal behavior monitoring techniques, especially the ones that use machine learning (ML) techniques. Section 3 describes the data gathering methodology and ML process performed in the scope present work, and Section 4 presents the data analysis and ML outcomes. Section 5 discusses the results and Section 6 concludes the paper, presenting the most relevant conclusions and pointing out future work.

## 2. Related Work on Behavior Classification Based on ML

The use of animal sensing technologies with precision livestock farming systems [5], integrated according to a typical architecture of the Internet of Things, has allowed the collection of representative data from the most diverse aspects of animal life. After being filtered, such data can be analyzed and important information related to animal activity [6,7], health [8,9,10] and feeding [11] can be extracted, ultimately becoming a very useful tool for improving the management efficiency in this area.

### 2.1. Animal Sensorization

Animal sensing based on wearable sensors has been used in numerous works [1,2], mostly through devices placed on the neck [2] and operated by a microcontroller. Konka et al. [12] studied the ability to detect ruminating and eating behavior in cows, comparing data from a collar and a device on the jaw. Although it is accepted that the device in the jaw, also with an accelerometer, can capture these behaviors more easily and with less noise, it is emphasized that it is also possible to classify them using the collar with an accuracy of over 90%. Giovanetti et al. [13] studied the detection of sheep grazing, rumination and resting behaviors, obtaining an accuracy also above 90%, thus showing the ability to detect behaviors of this kind with this type of device. Barwick et al. [14] studied the effectiveness of different placements for accelerometers (collar, leg and ear tag) to classify sheep grazing, standing and walking behaviors. They analyzed 14 features and correctly predicted 94%, 96% and 99% of grazing, standing and walking events, respectively, using ear acceleration values. Balasso et al. [15] studied the placement of an accelerometer in the left paralumbar fossa of cows, with high success in predicting postures (such as standing and lying down), but performing worse when predicting eating, walking, ruminating, and standing still behaviors, achieving results below 80%. This success can be explained by, in these postures, the animal continuing to move its head, which induces errors in other devices/models. Analogously, behaviors that are mostly associated with head movements are not detected by a device that aims to detect the position of the body.

Mansbridge et al. [16] compared the performance of collars and ear sensors with accelerometers and gyroscopes to detect grazing and ruminating behaviors in sheep. Despite the collar performance being superior, it was only by a relatively small margin, showing the ability of sensors in the ear to capture movements in a similar way to collars. Considering this, the fact that sensors in the ear are potentially more practical is highlighted, since they can be integrated into animal identification earrings (ear tag). Another study, also considering sheep [9], aimed to differentiate normal behavior from that of (simulated) lame animals. Due to the forced difficulty in walking, the head movements become more accentuated, allowing the respective accelerations to be captured more easily, evidencing again the ability of these devices to allow the detection of the condition prematurely. In the same study, performance with a collar (and a device on the leg) is compared to the ear device, and it is noted that accentuated head movements are “damped” by the location (neck), with more errors in the classification of behaviors with lameness. This resulted in a better performance of the device in the ear, in relation to the others.

There is also reference to the simultaneous use of devices. In a particular case, the use of a device on the leg synchronized with a collar is explored [17]. The study claims to have achieved better accuracy values compared to other similar works. This is because, regarding the postures and behaviors studied, the collar better reflects the movements of the mandible when the animal (in this case cows) eats, while the device on the leg better classifies lying down and walking, thus covering the “weaknesses” between the devices. However, negative aspects are pointed out in this type of approach, since it implies more costs in general, as well as other disadvantages, both for the animal and the handler [15]. This study highlights the evident difference in performance between animals that are used to the use of both devices in relation to those that are not.

Over the last few years, the authors of the present paper have been focusing on the development of an animal monitoring collar, first with the aim of conditioning postural behavior and preventing the animals from feeding on leaves and grapes and thus being able to thin out vines [18]. In this first version, in addition to an accelerometer, the collar had a distance sensor using ultrasound in order to compensate for the slopes of the vines in the accelerometer values, typical of the Douro region, Portugal. In tests conducted outside this environment [19], the information provided by the distance sensor never proved to be a relevant feature, justifying why it ended up being excluded from later versions of the collar. Subsequently, the collar was used to analyze the nocturnal activity of the sheep [20], as well as the goat kidding process [21], without using the data collected by the ultrasound sensor [6].

While devices equipped with 3-axis accelerometers are the most used type of approach in the context of animal monitoring, other alternatives can be of interest. As such, it is important to point out the possibility of using other sensors to monitor the welfare of the animals. Of these, it is possible to highlight heart rate and body temperature sensors (which may indicate a health problem), as an indirect method that serves as an indicator for stress and agitation levels [8]. Humidity detectors can be added to a sensory platform, as they may play a role in influencing the behavior and metabolism of animals [22].

### 2.2. ML Techinques for Animal Behaviour Classification

ML [23] is a branch of the field of Artificial Intelligence (AI), which seeks to use data and algorithms to imitate the way humans learn, gradually obtaining better results [24]. In the scope of animal monitoring, it is often used in the context of supervised learning [2], as it allows for the automation of this process. Here, algorithms are trained to predict outcomes based on labeled data, trying to make a mapping between the input and output.

As such, in a first stage, associated with the data collection based on sensors, a process of direct observation is normally implied, either in person, or through video recording, to carry out a subsequent labelling of data, based on the point in time a given behavior occurred.

Considering that data based in wearable sensors is numerical by nature, one of the most used algorithms are Decision Trees, or a more complex variation, Random Forests. Decision Trees perform well with numerical data, which translates into favorable learning capabilities. In addition, they are characterized by their legibility and simplicity, translating into the fact that they are computationally less intensive, which is of extreme importance when applied in microprocessors, adding to the fact that it allows to maximize the lifetime of the battery supporting the device. These aspects are highlighted by Kamminga et al. [25] as important criteria to be considered when selecting a ML algorithm for use in devices of this nature.

## 3. Materials and Methods

The present study consisted of creating a new dataset by monitoring three animals through a monitoring collar and the recording of video images. Specifically, it included a supervised learning process where an ethogram with five behaviors was defined, (namely Eating, Walking, Standing, Lying, and Ruminating). Through the recording and annotation of video images, the learning process was carried out and the ML results were analyzed. The most impactful features in the behavior learning process were analyzed (considering both the scenarios of including and omitting gyroscope data), and the complexity of the decision trees obtained was also assessed. Based on the chosen set of sensors, a learning model was subsequently established and the learning outcomes were analyzed. Finally, considering that the sensed data used for the learning model was based on adult animals, its behavior prediction capability was analyzed in a scenario featuring animals of a younger age.

### 3.1. Animal Monitoring Platform

The monitoring platform developed by iFarmTec (http://ifarmtec.pt/produtos.html, accessed on 19 December 2022), consists of a collar (Figure 1a) equipped with inertial sensors, integrated via Bluetooth communication with an Android application (Figure 1b) that centralizes the telemetry records and video images. Collar sensors include an InvenSense ICM-2948 IMU that senses a 3-axis gyroscope, 3-axis accelerometer and 3-axis compass, and its Nordic Semiconductor NRF52 includes a thermometer.

The learning firmware used in the experiments performs a 20 Hz sensor sampling and streams monitoring information through Bluetooth Low Energy (BLE) notifications to a BLE central Android application. It contains a set of sensors under evaluation, namely an accelerometer, a gyroscope, a magnetometer, and a thermometer. Additionally, it includes a Near Field Communications (NFC) sensor that acts as a power switch and allows the triggering of an NFC Peer-to-Peer (P2P) process that includes Bluetooth pairing between the BLE peripheral (collar) and BLE central (Android application).

Thus, after establishing a connection between the mobile device and the device on the animal, it was possible to use the application to both record video about the animal’s behavior and receive synchronized data from the sensors. Both the video and the sensorial data would then be stored in real time in separate files on the mobile device’s internal storage.

### 3.2. Monitored Animals and Monitoring Setup

Three animals were used in this study. The focus was on two adult female sheep (*Ovis aries*) of the “Serra da Estrela” autochthonous breed, aged between 3–4 years and 2.5 years, respectively. The third animal consisted of a male lamb four months of age, used to compare its behavior with the adults and to verify the generalizing ability of the algorithms developed with adult data to predict younger animal behavior.

Considering that the adults were the focal point of the study, more data were collected about the sheep than the lamb. As such, regarding the sheep, data retrieval happened during five non-continuous days. Throughout the days, it became possible to get progressively closer to the animals as their familiarity would increase, something that was also noticed in other works [26,27].

Regarding the remaining days, they were not continuous as to allow for the transferring of the data from the mobile devices to a computer and for a posterior pre-analysis of the data, looking to see the most predominant behaviors, at what time they occurred and which ones were the least captured.

Considering the behaviors themselves, they can be synthesized in the ethogram described in Table 1 and they include the following behaviors: Eating, Walking, Standing, Lying, and Ruminating.

Every recording took place in the afternoon, between June and July 2022, with the first day starting at around 5 p.m. However, a lack in the variety of behaviors was noticed, which lead to a successive delay on the start of the recording seasons on each of the next days as to bypass the effect of the hot summer weather. Useful data were collected from the two adult sheep over the span of four days. For this task, two mobile phones were used, so that each collar/animal was assigned to a specific mobile device throughout the whole gathering time. After each recording season those files were transferred to a computer for further analysis and processing. This includes the merging of the Comma-Separated Values (CSV) files of each animal into a single, unified one, for the day. Following this aggregation, a labelling task was performed, to create a dataset fit for supervised learning.

### 3.3. Pre-Processing and Data Cleaning

After having the data structured and having done a preemptive analysis, a cleaning of that same data was necessary. This process can be divided into a set of smaller steps. Duplicates were removed based on the timestamp that originated on the collar. Since no time value should be repeated (and have an increment of about 1/20 s from the previous entry), when this condition was not met the value on the features were also the same, further validating this condition for removal. Outliers were removed as well during data cleaning to avoid problems in statistical procedures, easing the machine learning processes. Due to data cleaning, the initial 459,750 obtained entries were reduced to 446,043 (an approximate 3% reduction).

The set of features available in the dataset, resumed in Table 2, included the acceleration values along the *x/y/z* axis, the gyroscopic values along the *x/y/z* axis, the dynamic acceleration value along the *x/y/z* axis [7], and pitch and roll angles.

Considering the resulting (adult) dataset, a split of 75/25 was taken for training and testing, respectively. Moreover, the split was performed under a stratified strategy, so that the splitting of data was proportional to class distribution.

### 3.4. Algorithm Choice and Model Evalutation

Considering the advantages of Decision Trees, it was the algorithm of choice for the learning task. Therefore, the depth of the tree determines the ability to specify through sub-conditions the features’ values that represent each class to be learned, and therefore increasing the tree depth increases the accuracy of the learning process. Consequently, the increase in depth leads to an exponential increase in the number of leaves, which has a strong impact on the code to be implemented in the classifier, with a special expression in microprocessor-implemented classifiers used on wearable devices.

Regarding the way in which model performance is evaluated, accuracy is often used. Notwithstanding, animal behavior is naturally erratic, which causes an imbalance in class distribution. For this reason, Balanced Accuracy can be used, as it accounts for different class distribution. An extremely useful tool in the analysis of supervised learning outcomes is the confusion matrix, which represents in each row the instances in an actual class, while each column represents the instances in a predicted class, or vice versa. A high number of errors in the prediction between two classes is typically due to the similarity of the classes’ characteristics, to a poor sensing process, or even to the lack of sample representativeness of a class in the dataset used in the learning process. Additionally, other metrics allow to confer the performance of each behavior individually, namely Precision and Recall. In this context, Precision shows how much a model can be trusted when predicting a given behavior, while Recall expresses the capability of a model to find all the entries of a given behavior. In addition, F1-Score was also used, as it can be interpreted as the weighted average between Precision and Recall.

## 4. Results

To find a model capable of predicting animal behavior, two main metrics were used to evaluate the performance of the developed models: (i) the balanced accuracy; since the dataset was not balanced, this variation of accuracy was employed, as it accounts for different class distribution and measures how much a model correctly predicts the entire dataset; and (ii) the number of leaves was primarily used to compare the complexity of different models. In other words, the bigger this number is, the more complex a tree usually is. Note that unless stated otherwise, the results shown refer to the testing set.

The effect of a decision tree’s (maximum) depth was one of the parameters analyzed. Figure 2 graphically highlights the results obtained from this experiment.

Observing the graphic related to balanced accuracy (Figure 2a), it is possible to see how the model’s performance tends to rise with the tree depth increase. Notwithstanding, after a certain depth, testing accuracy growth stops following the growth of the training set, indication of overfit. Regarding the number of leaves (Figure 2b), it is possible to see an exponential growth based on the tree’s depth.

While other hyperparameters were experimented with, the results were not considered relevant or conclusive to be considered in the scope of this paper. Notwithstanding, considering the tests performed and the set of all the features, an initial best model was reached, whose learning results can be consulted in Table 3, along with a confusion matrix illustrated in Figure 3.

Attending to Table 3, it is possible to verify that both the values of training and testing accuracy are around 92%, meaning the model could accurately predict most classes. This value was in line with similar works and therefore is highlighted positively. Moreover, it is important to note how the number of leaves (448) is significantly lower than the numbers obtained during the study of depth influence (Figure 2, right), where no tuning was performed, highlighting the improvement of this model.

The analysis of the confusion matrix shows two problems, both related to the small sample size. First, there was a huge deficit of W-classified samples, which typically has a consequent learning deficit for the W behavior. Second, the model often confused the E behavior with S and W behavior. The later problem had to do with the dynamic similarities existing between the states and can be mitigated through the extension of the monitoring period. The model served as a baseline for the following experiments, in which the effect of features/sensors was considered.

To determine which features were more impactful for predicting the behaviors, the contribution of each feature for the model was also assessed based on the effect that each feature had on the model’s prediction, e.g., how much each feature was used on the tree, and its results are described in Table 4. Moreover, the closer to the root a given feature/split is, the more important it is, since it has an influence on the splitting of a larger amount of data. Its analysis allows us to perceive that the most important feature is the Temperature (TMP), followed by Acc(*x*) and Pitch. Considering the high correlation between the pitch and the acceleration values, the enormous impact of the features provided by the accelerometer for the accuracy of the developed learning process was recognized.

To assess the impact each sensor had on the performance of the models, a specific study was later carried out. As such, different models were trained based on distinct feature sets. Considering the previous results, the accelerometer was taken as base model, and a separation was defined within the features of this sensor. The first subset, StaticAccFeatures, consists solely of the features directly extracted from the sensor, i.e., the values of acceleration on each of the three axes (*x*, *y*, *z*). The second subset, AccFeatures, in addition to the former, also contains the values of dynamic acceleration for each axis. Finally, InertFeatures contains not only the features of the previous subset, but also the values of Pitch and Roll. These two features can be calculated indirectly, based on trigonometric calculations over the values of the accelerations. Hereupon, each of these subsets translates into additional overhead that can happen on a microcontroller, so trying to minimize the number of calculated features may prove beneficial for a device’s performance. Table 5. summarizes each of the feature subsets.

The performance of each set of features was then evaluated, based on two metrics: the balanced accuracy and the tree complexity associated with the increase of the tree depth. The first one measured the balanced accuracies of each model, and the results are illustrated in Figure 4. It is important to highlight that AllFeatures corresponds to the set of all the features, while “Thermometer Contribution” and “Gyro Contribution” reveal the difference in performance based on the impact each sensor has on the models without it.

By observing Figure 4, it is possible to conclude that specific increments in the features are beneficial to the performance of the models. In other words, while there is an increment in balanced accuracy from the StaticAccFeatures model to the AccFeatures and InertFeatures models, the difference between the latter two is negligible, suggesting that Pitch and Roll do not add value to the learning process. Likewise, the model InertFeatures + TMP performs similarly to the one with AllFeatures, indicating that Temperature plays an important role in the performance of the model, in contrast to Gyroscopic features (the difference between these two models).

The second metric analyzed was the complexity of the tree, based on the number of leaves. The results are shown in Figure 5, which allows a differentiation between the models based on complexity. Here, it is important to mention how every subset of acceleration models (StaticAccFeatures/AccFeatures/InertFeatures) presents a similar number of leaves.

The contribution of each of the sensors to the learning process could thus be isolated, which, together with the cost of each of them, allows an informed decision regarding the most appropriate implementation of the collar. Table 6 synthetizes the results obtained for the best performing model as a function of each removed sensor, along with the financial impact of each choice, compared to the model with all the sensors.

Comparing the results of Table 6, the benefit of removing the Gyroscope can be highlighted considering multiple aspects. Albeit by a small margin, there is a slight improvement in the testing balanced accuracy, and a reduction in the number of leaves comparatively to the AllFeatures model. Additionally, this sensor is relatively expensive compared to the others.

Based on the analysis of the cost and benefit of each of the sensor sets, it was decided to remove the gyroscope and to ignore the contribution of the associated features (Gyr(*x*, *y*, *z*)) to the learning outcome. Table 7 summarizes the learning metrics, and Figure 6 defines the confusion matrix obtained.

Note that the Support (as shown in Table 7 and Table 6) refers to the number of data entries of the respective state, and it illustrates how each state was represented in the testing set.

Another hypothesis tested by this study was the capability of a simple model to predict the behaviors of an animal of the same species but in an early development phase, namely, a growing lamb. The testing set consisted of the whole lamb’s dataset, as it was considerably smaller than the adult’s counterpart. As such, the InertFeatures + TMP model was used and a balanced accuracy of 0.57 was obtained. Table 8 and the confusion matrix illustrated in Figure 7 show the calculated results.

Here, it is relevant to draw attention to the fact that, while the overall performance of the model was sub-optimal, some behaviors performed significantly better than others. Specifically, E has a solid metric of both precision and recall, resulting in an equally good f1-score. State L, on the other hand, when classified, had a tremendous performance, nearly being perfectly classified (due to a precision of 0.98), even though the model failed to find most of L state entries, as shown by its low recall.

## 5. Discussion

The analysis of the most impactful features in the learning process allowed us to perceive that features associated with the accelerometer were dominant, both in terms of linear accelerations (i.e., Acc(*x*, *y*, *z*)) and the features dependent on them (i.e., Pitch, Roll). Regarding the temperature, although the inclusion of the thermometer was not motivated by its importance in behavior detection, the results obtained suggest that this feature has an important role in improving predictive power. It was a surprising and interesting result, with a possible explanation having to do with the greater transfer of heat to the thermometer when the animal lies down, which helps to identify this behavior.

The analysis of the model’s performance illustrated in Figure 2, with the increase of tree depth, revealed an important point: the depth increase influences the balanced accuracy of the model, since as the value of depth increases, so does the predictive capacity of the tree. However, this growth was neither linear nor indefinitely beneficial. Upon reaching a certain point, the testing accuracy stops following the training growth, indicating that the tree starts to overfit. Moreover, the number of leaves (a measure of tree complexity) expands exponentially based on the tree’s depth. This value was of particular importance for two reasons. On the one hand, it is an indicator of the computational demand that the model has on the classification algorithm processor, which, in this case, is a microprocessor with reduced processing capability and memory capacity, in order to keep both the production costs and energy consumption low. On the other hand, while an increment of this value might be beneficial for a model’s overall performance, if unchecked it can result in a situation of “diminishing returns” in which the increase in accuracy stops following the increase in complexity. A high number of leaves can be an indicator of how specific the tree is getting towards data, e.g., whether it the model is getting high variance or not. This exponential growth is expected to stop after a certain depth, as nodes start to get pure, i.e., leaves begin to contain data from a single class only, putting branch growth on hold.

The analysis of the contribution of each of the sensors to the global performance of the learning process allowed us to identify a huge impact of the accelerometer data on learning, both due to the linear acceleration features it provides and the derived features. Moreover, while the models with AllFeatures and InertFeatures + TMP presented a similar value of balanced accuracy (Table 6), it is possible to see how the exclusion of gyroscopic features is beneficial, with a small increment in the balanced accuracy of around 0.3%. While this increment might appear negligible, it is important to attend to the fact that it also translated into a reduction of 27 leaves, suggesting that gyroscopic features not only are non-informative but also add noise to the model, and with that noise, unnecessary complexity. It was also possible to verify a slight contribution provided by the thermometer, almost consistent, regardless of the depth of the decision tree (Figure 6). In contrast with the results obtained for the thermometer, the gyroscopic-related features were ranked very low in the feature ranking phase. Thus, the gyroscope appeared as a sensor capable of being removed without negatively impacting the model. Furthermore, when considering the sensor cost, another reason arises to justify its removal.

The evaluation of the learning results obtained without resorting to the gyroscope made it possible to consider the feasibility of creating an automatic behavior classification mechanism, even though some of the behaviors are still classified with very low accuracies. Since the present work is still exploratory and has the concrete objective of evaluating the set of necessary sensors, the set of monitored animals was very low, and the monitoring period was very short. Therefore, the number of samples related to states is very small and little representative of the diversity of animal anatomy and behavior, which translates into a low capacity to learn these states. Once the hardware with the sensors determined in this work has been produced, a new learning process will have to be conducted with more animals, of different breeds and during different times of the year.

Regarding the prediction of a lamb’s behavior based on the training of an adult’s dataset, results suggest that such a task is not possible. With a balanced accuracy of 57%, the model cannot accurately predict the range of behaviors the lamb displays. Still, it is important to highlight how Eating performs noticeably well, based on the confusion matrix of Figure 7. This might be justified by the fact that the posture taken by the lamb during this behavior is very identical to the sheep. Likewise, the model has a lot of confidence classifying Lying, possibly for the same reasons, although it fails at finding all Lying entries. This poor performance, as well as the performance of the remaining behaviors, might be justified by the fact that the young animal is more active and its movement more dynamic compared to the adult counterparts, something typical of the animal’s youth.

## 6. Conclusions

Autonomous monitoring presents itself as an answer to many issues, with many benefits but also several disadvantages. Selecting which kind of sensors to use and techniques to employ dictates the costs of producing a system for this problem, how much gain will be obtained from its employment, and how hard it would be to maintain such a system.

In the present study, the impact of a set of sensors on the detection of animal behavior through a supervised learning process was evaluated and the most impactful features present in the dataset in the learning process were identified. Results confirmed the importance of the features provided by the accelerometer for the accuracy of the learning process and indicated an interesting impact of the data provided by the thermometer on the learning outcomes. Conversely, the data provided by the gyroscope in the learning process did not improve the learning outcomes, having proved to be a redundant source of information to the information provided by the accelerometer, and the removal of the gyroscope ends up being beneficial, albeit by a small margin. This conclusion is of particular importance since gyroscopes are expensive, and the possibility of removing this sensor without affecting the performance significantly is valuable.

This work was subject to limitations that future iterations should consider and overcome. A small set of sheep was evaluated. With only two (adult) sheep available to be evaluated and compared, the results are limited. Thus, conclusions about the power to generalize to other animals (within the same growing phase or not) are also limited. For this type of experiment, a larger number of animals would confer higher certainty on the results obtained. Similarly, a larger diversity of data from lambs would confirm that their behaviors cannot be predicted with models trained for adults. A future line of work that should be followed is the comparison of activity dynamism and behavior between animals of different age groups.

Further studying the temperature could be valuable. The scope in which this feature was experimented was limited, due to harsh logistics and necessary work. Thus, data was retrieved during the summer season, in the afternoon. Extending the observation period to a whole day could be advantageous to seek the collection of a more diverse set of data. Likewise, expanding this gathering to a longer period could be advantageous to observe the variation and influence of temperature throughout the year. Additionally, other factors, such as the climate and animals in heat, could be considered. With that, some mechanism to normalize temperature as a function of the time of the year could be necessary, to improve generalizing power.

## Figures and Tables

**Figure 1 animals-13-00120-f001:**
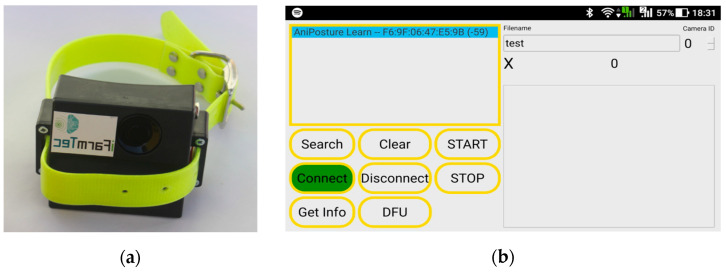
Monitoring platform: (**a**) monitoring collar, (**b**) android app that gathers collar data and video records animal behavior.

**Figure 2 animals-13-00120-f002:**
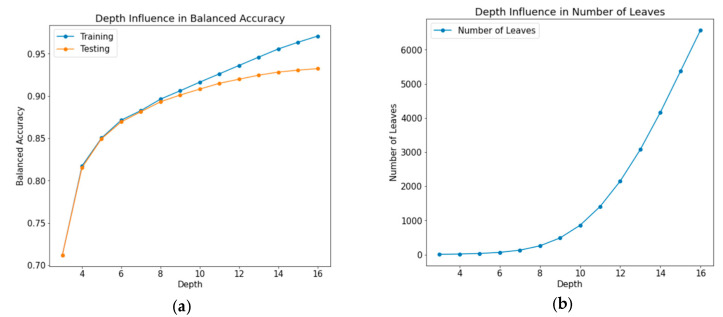
Model’s performance as function of depth: (**a**) tree depth influence in balanced accuracy, and (**b**) tree depth influence in number of leaves.

**Figure 3 animals-13-00120-f003:**
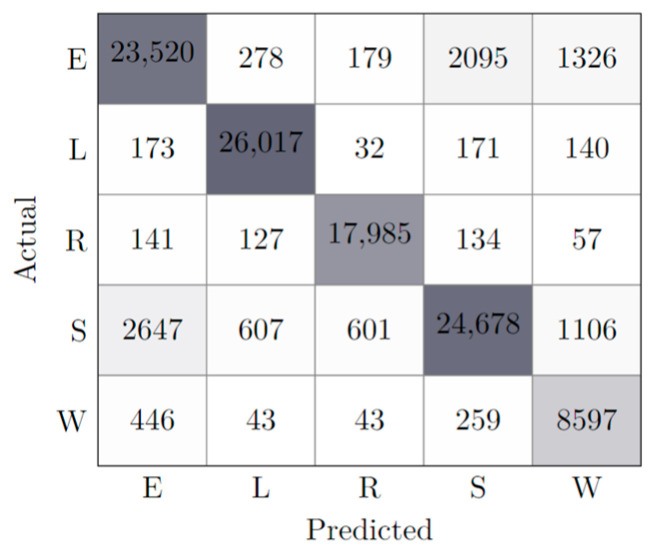
Confusion matrix of best model (AllFeatures). The different cell colors represent the classification accuracy; the more accurate the classification of a state, the darker the cell is represented.

**Figure 4 animals-13-00120-f004:**
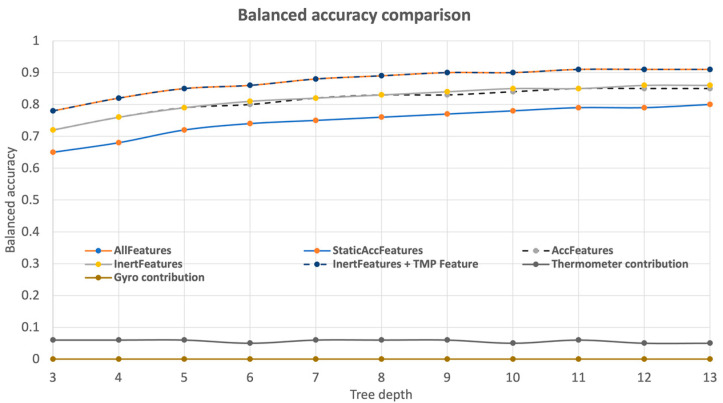
Balanced accuracy evolution with tree depth.

**Figure 5 animals-13-00120-f005:**
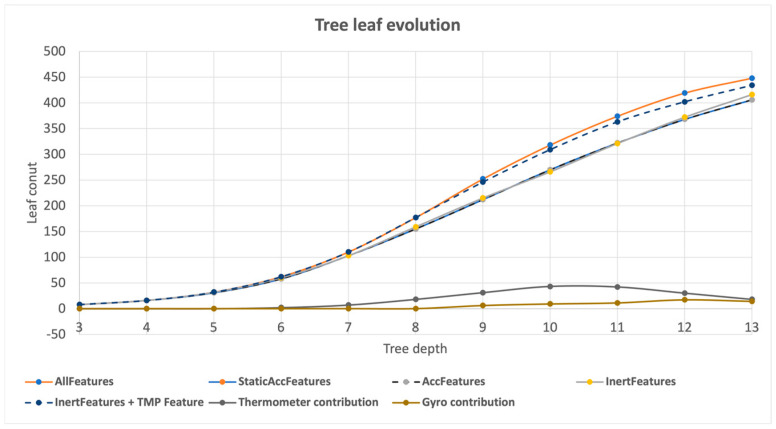
Tree leaf count evolution with tree depth.

**Figure 6 animals-13-00120-f006:**
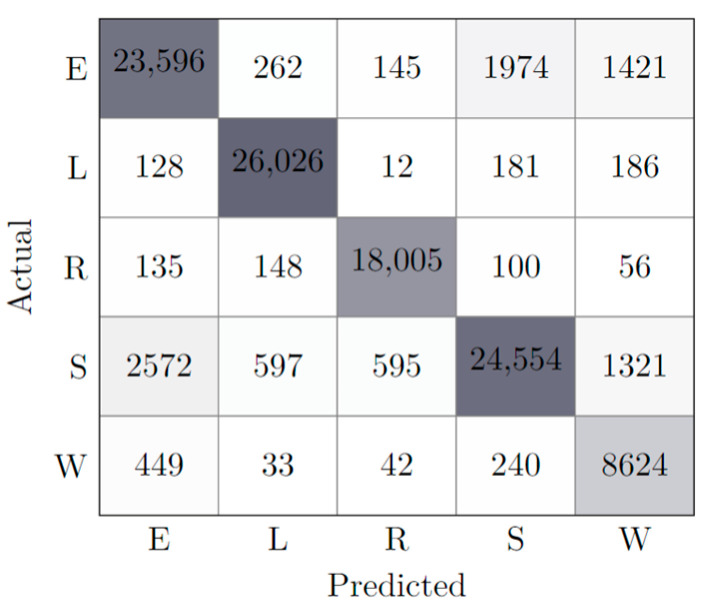
Confusion matrix of the best model (InertFeatures + TMP). The different cell colors represent the classification accuracy; the more accurate the classification of a state, the darker the cell is represented.

**Figure 7 animals-13-00120-f007:**
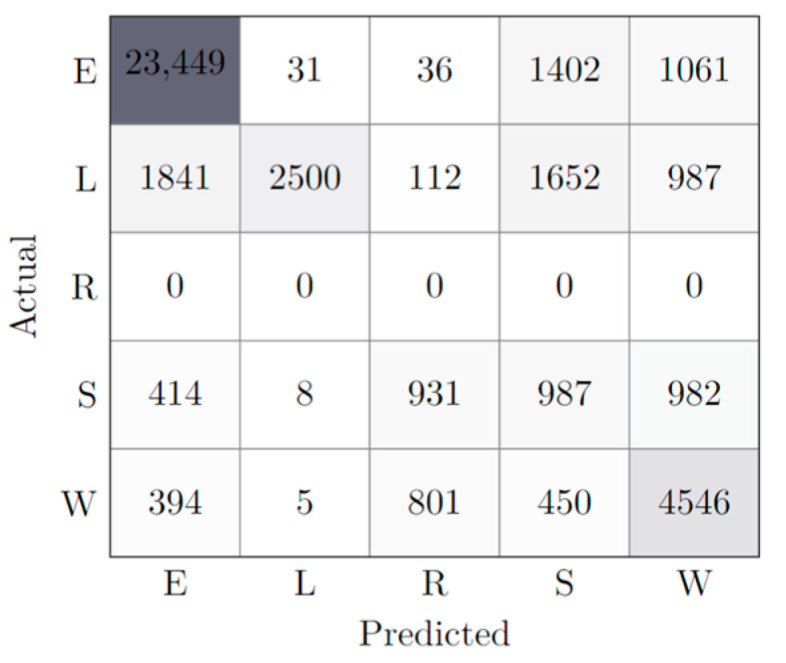
Performance of Best Model (InertFeatures + TMP) against Lamb. The different cell colors represent the classification accuracy; the more accurate the classification of a state, the darker the cell is represented.

**Table 1 animals-13-00120-t001:** Ethogram of captured behaviors.

State	Description
E—Eating	Eating grass. Small steps and head movements while having the muzzle down, natural from this process, still count for this classification. Chewing with the head slightly up is still accounted for.
W—Walking	Moving for a considerable amount of time, either at a slow or a fast pace.
S—Standing	Stationary behavior. Neck movements to look around are accounted for.
L—Lying	Lying down with head lowered or only slightly raised.
R—Ruminating	The act of regurgitating food from the stomach back into the mouth for proper chewing. Can be done while standing or lying (but exclusive of those behaviors)

**Table 2 animals-13-00120-t002:** Feature description.

Feature	Description
Acc(*x*, *y*, *z*)	Acceleration value along the *x/y/z* axis
Gyr(*x*, *y*, *z*)	Gyroscopic value along the *x/y/z* axis
D_Acc(*x*, *y*, *z*)	Dynamic acceleration value along the *x/y/z* axis
TMP	Temperature measured at microprocessor board
Roll/Pitch	Roll/Pitch angles

**Table 3 animals-13-00120-t003:** Performance of best model (AllFeatures).

Parameter	Value
Balanced training accuracy	0.919
Balanced testing accuracy	0.912
Leaves	448

**Table 4 animals-13-00120-t004:** Feature importance of the best model.

Feature	Value
TMP	0.270
Acc(*x*)	0.191
Pitch	0.184
D_Acc(*x*)	0.176
D_Acc(*y*)	0.055
Roll	0.038
D_Acc(*z*)	0.031
Acc(*y*)	0.020
Gyr(*y*)	0.013
Acc(*z*)	0.010
Gyr(*z*)	0.008
Gyr(*x*)	0.003

**Table 5 animals-13-00120-t005:** Feature set used.

Feature Set	Description
StaticAccFeatures	Acc(*x*, *y*, *z*)
AccFeatures	Acc(*x*, *y*, *z*) + D_Acc(*x*, *y*, *z*)
InertFeatures	Acc(*x*, *y*, *z*) + D_Acc(*x*, *y*, *z*) + Pitch and Roll
InertFeatures + TMP	Acc(*x*, *y*, *z*) + D_Acc(*x*, *y*, *z*) + Pitch and Roll + Thermometer
AllFeatures	Acc(*x*, *y*, *z*) + D_Acc(*x*, *y*, *z*) + Pitch and Roll + Thermometer + Gyro(*x*, *y*, *z*)

**Table 6 animals-13-00120-t006:** Balanced accuracies by sensor.

Removed Sensor	BAcc Training	BAcc Testing	Leaves	Saving
All sensors present (AllFeatures)	0.919	0.912	448	---
No accelerometer (no InertFeatures)	−0.014	−0.010	−114	$1.6
No gyroscope (InertFeatures + TMP)	0.000	+0.003	−27	$7–1.6
No thermometer	0.038	−0.037	−34	$0
No angular features (Pitch and Roll)	−0.000	+0.001	−48	$0

**Table 7 animals-13-00120-t007:** Scores of best model (InertFeatures + TMP).

Metric	Precision	Recall	F1 Score	Support
E	0.88	0.86	0.87	27,398
L	0.96	0.98	0.97	26,533
R	0.96	0.97	0.97	18,444
S	0.91	0.84	0.87	29,639
W	0.75	0.92	0.83	9388
Average	0.89	0.91	0.90	111,402

**Table 8 animals-13-00120-t008:** Scores of best model (InertFeatures + TMP) against Lamb.

Metric	Precision	Recall	F1 Score	Support
E	0.90	0.90	0.90	25,979
L	0.98	0.35	0.52	7092
R	0.00	0.00	0.00	0
S	0.22	0.30	0.25	3322
W	0.60	0.73	0.66	6196
Average	0.54	0.46	0.47	42,589

## Data Availability

Not applicable.

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
