# Peer review of "On the Development of a Wearable Animal Monitor"

_animals, 2022, doi:10.3390/ani13010120_

Round 1

Reviewer 1 Report

The title of this paper mentions that the contribution is a wearable animal monitor. However, the animal monitor is or was on the market and the authors are processing the data obtained for 3 lambs during June and July. 

First, change the title, this is a data processing article, not a hardware development. 

In the same line, include more information about the product, The IFarmTec web page is not working. Include information about battery duration, price on the market, durability, etc.

In the abstract, the authors describe the activity of pastoralists as solitary and unpopular. However, according to the methodology, pastoralists should connect their smartphone to the monitor collar and store the information on a computer. 

In this reviewer's opinion, the data coming from the animal is the main drawback of this paper. As the authors claim is difficult to work with animals, and collect data. In this process, many variables can affect the quality of the information. Also, this part will require a more intense cleaning process.

This reviewer recommends improving the methodology. Include a flow diagram, and consider more animals, extend the period of monitoring. Also, add some pictures to clarify the data-gathering process. 

If this part is not convincing and well described, the results lack certainty. 

The conclusions are very extensive.

Avoid typos, for example, line 236 ----- 17 pm

it should be 5:00 pm

Author Response

The title of this paper mentions that the contribution is a wearable animal monitor. However, the animal monitor is or was on the market and the authors are processing the data obtained for 3 lambs during June and July.

We thank the reviewer for effort in reviewing the paper and for his valuable comments.

 There is a misconception regarding the product that is on the market and what was used for the work in our paper. The product on the market consists of a behavior conditioner, to allow sheep to graze in the viticultural space. For the work in our paper, we are only reusing the outer box, and only temporarily.

First, change the title, this is a data processing article, not a hardware development.

The paper discusses the process of evaluating the performance of a collar version and evaluates its predicting behavior accuracy of a candidate version of the hardware. The work documented allowed to decide the IMO to be disconsidered in next version of the hardware. The collar is an embedded system controlled by a microprocessor and as we documented in the paper, the collar firmware is to be developed based on a machine learning process (decision trees) that are directly translated to a set of algorithm conditions. In summary: we are actually in the middle of the system development process, using field data to support the development of the device.

In the same line, include more information about the product, The IFarmTec web page is not working. Include information about battery duration, price on the market, durability, etc.

The animal monitor equipment is still under development, and we do not know its durability, its battery autonomy nor its market price, mainly because we did not yet close its hardware nor its software, and both of them have a strong impact on power consumption and on cost. We can, however, report that our non-fully tuned operating set has a 3-month autonomy, but we cannot predict final values. iFarmtec website had a down episode, but it is up now.

In the abstract, the authors describe the activity of pastoralists as solitary and unpopular. However, according to the methodology, pastoralists should connect their smartphone to the monitor collar and store the information on a computer.

Collars communicate using a Bluetooth Low Energy (BLE) interface, either to a smartphone, or to a dedicated gateway that will transfer the monitoring information to a cloud application hosted in the Internet. Collar implements temporary storage of information until BLE connection is achieved, so it is not necessary to have continuous BLE connection. Additionally, we hope that our new technology, and the use of smartphones, can make herding less monotonous and lonely.

In this reviewer's opinion, the data coming from the animal is the main drawback of this paper. As the authors claim is difficult to work with animals, and collect data. In this process, many variables can affect the quality of the information. Also, this part will require a more intense cleaning process.

The supervised learning process that allows extracting the conditions from the decision tree, requires proximity between the person filming and recording the collar data. As pointed out in the paper, such activity is complex. The process requires that many hours of supervision are spent so that the records in which the animal's behavior is visible can be used, without human interference. After the learning process, the monitoring of the animals takes place without human presence, and the data is downloaded from the collar through a gateway, with no human intervention.

This reviewer recommends improving the methodology. Include a flow diagram, and consider more animals, extend the period of monitoring. Also, add some pictures to clarify the data-gathering process.

The work documented in this paper consisted of assessing the importance of a set of sensors in the computational learning process, with the aim of creating a new animal monitoring device. The work was exhaustive in terms of analysis of the learning results, and in the analysis of the impact of each sensor, but not in terms of monitoring, nor in terms of obtaining an algorithm for classifying animal behavior. The process of learning and establishing the behavior classification algorithm will be repeated later with more animals, for a longer period of time and covering different seasons of the year and different periods of the day. Despite these limitations and the work that will be done in the future, the work documented here is not without merit, since it contributes to demonstrating that the most expensive sensor (the gyroscope) is completely redundant, and that the thermometer included for other purposes has a role determinant in the detection of some states.

If this part is not convincing and well described, the results lack certainty.

Machine learning results are clearly poor, and cannot be used as a behavior classification method, the valid results the paper presents have to do with the certainty that gyro is not relevant, since the accelerometer will perform the same sensor function, and that the thermometer offers a very relevant feature.

The conclusions are very extensive.

We agree with the reviewer that the conclusions are extensive, and we summarized them.

Avoid typos, for example, line 236 ----- 17 pm

We underwent a rigorous revision of the text and fixed them.

Reviewer 2 Report

General comment

The present manuscript titled “on the development of a wearable animal monitor” evaluated the potential of using low-cost sensors with effective modeling in machine learning. The finding that animal monitoring can successfully be done without using expensive sensors like gyroscope is worth sharing with the interested readers. The manuscript was enjoyable to read and easy to follow. An animal scientist with a small background in machine learning can easily demystify the processes and models applied in the study. I must appreciate the authors for this. The models tested in the study would be a great help for the technology researchers to develop cost effective devices for animal monitoring. Some of the suggestions are listed below to improve the quality of the manuscript.

Please add the summary of 200 words at the start. It is the journal format to have a summary at the start.

The abstract was more than 300 words. Please reduce it to 250 words. The sentences from line 14-17 and 21-24 can be removed without losing any important information.

Line 36: The introduction section needs some major improvement. Please follow the comments below.

Line 63-75: This paragraph described the methodology of the study that may come in materials and methods section.

Line 76-81: Please delete this paragraph.

Line 84-91: Delete this repetitive text.  

Line 94-153: Summarize this information into one paragraph of 5 or more lines. It could be the second paragraph of the introduction section. There is no need to have separate subheadings about the study background and review of literature.

Line 154-187: The details of machine learning and how it functions might not be the scope of the manuscript and should be removed. The description of decision tree, depth, leaves, and balanced accuracy should be presented in the Materials and Methods section.

There must be a paragraph at the end of the introduction section highlighting the objective and hypothesis of the study.

Line 189: The manuscript had sufficient information to repeat the study. However, the past tense should be used while describing the methods followed, sensors used, and models applied.

Results: The results were simple to understand. Please use past tense where applied while describing the findings.

Table 6: please add a row to present the values of the best model for quick reference.

Line 360: Was the best model the one with no accelerometer or the one with all features?

Line 370: Use the verbs like show or present instead of expose.

The discussion section was well explained.

Line 435-445: Please delete these lines from the conclusion section.

Line 446-469: Please concise the conclusion into one paragraph with a couple of take away points something like; the predictive models were effective to monitor sheep behavior without using gyroscope.  

Line 470-476: This section could have a subheading of limitations of the study.

Spellings error: line 161, 238, 253, 336.

Author Response

The present manuscript titled “on the development of a wearable animal monitor” evaluated the potential of using low-cost sensors with effective modeling in machine learning. The finding that animal monitoring can successfully be done without using expensive sensors like gyroscope is worth sharing with the interested readers. The manuscript was enjoyable to read and easy to follow. An animal scientist with a small background in machine learning can easily demystify the processes and models applied in the study. I must appreciate the authors for this. The models tested in the study would be a great help for the technology researchers to develop cost effective devices for animal monitoring. Some of the suggestions are listed below to improve the quality of the manuscript.

We take the chance to thank the reviewer for his effort in reviewing the paper and for his valuable suggestions.

Please add the summary of 200 words at the start. It is the journal format to have a summary at the start.

We wrongly thought it was optional, and we included the summary in this revised version of the paper.

The abstract was more than 300 words. Please reduce it to 250 words. The sentences from line 14-17 and 21-24 can be removed without losing any important information.

It was our inattention, we've corrected it.

Line 36: The introduction section needs some major improvement. Please follow the comments below.

Line 63-75: This paragraph described the methodology of the study that may come in materials and methods section.

We transferred the text to 3. Materials section.

Line 76-81: Please delete this paragraph.

We are not sure if we clearly got the recommendation, but this is the typical paragraph that end the introduction sections

Line 84-91: Delete this repetitive text. 

We did it.

Line 94-153: Summarize this information into one paragraph of 5 or more lines. It could be the second paragraph of the introduction section. There is no need to have separate subheadings about the study background and review of literature.

We did it.

Line 154-187: The details of machine learning and how it functions might not be the scope of the manuscript and should be removed. The description of decision tree, depth, leaves, and balanced accuracy should be presented in the Materials and Methods section.

The collar is an autonomous wearable device composed of a set of sensors operated by a microprocessor. The development of the firmware that controls the device is being developed by an algorithm obtained thought a ML process. We think the ML content cannot be removed from the paper, since it will allow readers to replicate our experience and understand why some choices were made.

Regarding the second suggestion, a new subsection (3.4 Algorithm Choice and model Evalutation) was created for that purpose. This created an opportunity to both explicit the ML algorithm used and to explain some evaluation metrics not touched before.

There must be a paragraph at the end of the introduction section highlighting the objective and hypothesis of the study.

We did it.

Line 189: The manuscript had sufficient information to repeat the study. However, the past tense should be used while describing the methods followed, sensors used, and models applied.

To write

Results: The results were simple to understand. Please use past tense where applied while describing the findings.

The whole section (4. Results) was edited to use past tense.

Table 6: please add a row to present the values of the best model for quick reference.

We did it.

Line 360: Was the best model the one with no accelerometer or the one with all features?

The best model referred to AllFeatures model. We fully edited the paper in order to increase the understandability.

Line 370: Use the verbs like show or present instead of expose.

We did it.

The discussion section was well explained.

Perfect.

Line 435-445: Please delete these lines from the conclusion section.

We removed most of them.

Line 446-469: Please concise the conclusion into one paragraph with a couple of take away points something like; the predictive models were effective to monitor sheep behavior without using gyroscope. 

It was done.

Line 470-476: This section could have a subheading of limitations of the study.

Yes, but since we strongly summarized the conclusions, probably it does not make sense anymore.

Spellings error: line 161, 238, 253, 336.

We did fix them.

Reviewer 3 Report

This paper presents an interesting piece of research that further explores the use of sensors for behaviour observation in sheep. This concept has already been comprehensively studied in the literature, however the current paper adds additional value and originality to the overall body of work in this area. 

The paper is well written (though minor English and grammar corrections are required) and easy to follow. The discussion is light and would benefit from more robust discussion of the results. Furthermore, the materials and methods and results sections would benefit from some minor changes/additions to improve the clarity of the work; for example, some figures are difficult to read and it remains unclear whether all work was conducted on the test set and how the test set was developed. 

Specific comments are as follows:

Abstract & general comments

·      Some general rephrasing and grammar corrections are needed e.g., line 15-16 is not a clear sentence.

·      Please reword the sentence starting Line 16 – I would disagree that pastoralism is unpopular. It is a common method of farming in many parts of the world.

·      Please review all references and ensure they fit the journal’s requirements. E.g., An author’s first name does not need to be included (Line 110, 170 etc)

·      There is an inconsistency with the use of comma and period decimal points throughout. Please amend

Introduction

·      Giovanetti et al [15] was done on sheep (Line 100). It currently reads like this research was done on cattle. Please clarify. You might also like to include a short discussion about the difference in detection between the two species (e.g., inertial differences due to the size of cattle over sheep) – see  Barwick, J., Lamb, D., Dobos, R., Welch, M., & Trotter, M. (2018). Categorising sheep activity using a tri-axial accelerometer. Computers and Electronics in Agriculture, 145, 289-297.

·      Please specify which country the Douro region is in

·      You may wish to include further discussion around the welfare benefits of sensor use (Line 146-152) – see Fogarty, E. S., Swain, D. L., Cronin, G. M., & Trotter, M. (2019). A systematic review of the potential uses of on-animal sensors to monitor the welfare of sheep evaluated using the Five Domains Model as a framework. Animal Welfare, 28, 407-420. https://doi.org/10.7120/09627286.28.4.407

·      Please include further discussion as to why you have chosen to test the model on a lamb. Using the literature, explain what differences there are and why the model may perform differently on a younger animal. You haven’t yet provided sufficient background information as to why this particular research question has been included.

Materials and Methods

·      The access date in Footnote 1 is incorrect (page 4)

·      Remove reference to ‘both females’ (line 217). It is stated above.

·      Change 17pm to 1700 or 5pm (Line 235)

·      Change “spawn” to “span” (Line 238)

·      Spelling of “angles” (Line 258)

·      Please include information on the development of the ML algorithms here. What was your training/test split? How did you split the data? Please also include a description of precision, recall and f1score as you have done for balanced accuracy.

·      There has been a number of different features used in behaviour algorithm development in the literature. More details are required on the features used, parituclarly the dynamic acceleration of each axis (Line 319). Consider including an equation and/or a reference to another paper that provides this information.

Results

·      Figure 2 (and any others) should have the two figures referenced as (a) and (b) with a caption description that covers both

·      Figure 4 is difficult to read and interpret. Consider splitting out the graphs into a faceted graph or changing the colour scheme. I also cannot see the line for All features.

·      Table 6 is confusing to read. For example, in the row titled “No gyroscope”, the sensor cost is then listed as $7, which we know is the cost of the gyroscope from your introduction. This leads to confusion. Please amend the table so it is easier to comprehend. I suggest you also include the previous best model in this table for easier comparison.

·      Line 358. I suggest you use a different word than “notorious”. It is not a correct description. The increase in accuracy is marginal at best.

·      You may benefit from a different naming convention for your models and a Table that summarises them. It is difficult to determine which model is being discussed in the current paper.

·      The sentence from Line 375-377 either requires rewriting or further clarification as to what it means to have a high precision but low recall.

·      Table 7 & Table 8 – what does support refer to?

·      Could the author please confirm that all the results presented for the development of the model are for the test data set? Not the training set?

Discussion

·      Line 385-386. Is this suggestion of the heat transfer when lying the author’s hypothesis. Or is there supporting data for this? Further discussion is needed.

·      Please include further discussion as to why tree complexity is or is not beneficial in terms of algorithm development and use (Lines 387-397). You have not explored this fully and your paper would benefit from further discussion.

·      Please also provide further discussion as to the limitations of this work with regard to the small sample size (number of animals & data collected). It is briefly stated from Line 420, but further discussion of these limitations is necessary.

·      Please provide further discussion as to why the author’s believe the algorithm did not work on the lamb. You have briefly discussed this from Line 424 but further discussion would be beneficial.

Conclusion

·      Suggest you specify ‘autonomous’ monitoring or the use of sensors. Not just ‘monitoring’. Monitoring as a general concept can be performed by visual observation.

Author Response

This paper presents an interesting piece of research that further explores the use of sensors for behaviour observation in sheep. This concept has already been comprehensively studied in the literature, however the current paper adds additional value and originality to the overall body of work in this area.

We thank the reviewer for his effort in reviewing the paper and for his valuable comments.

The paper is well written (though minor English and grammar corrections are required) and easy to follow. The discussion is light and would benefit from more robust discussion of the results. Furthermore, the materials and methods and results sections would benefit from some minor changes/additions to improve the clarity of the work; for example, some figures are difficult to read and it remains unclear whether all work was conducted on the test set and how the test set was developed.

Specific comments are as follows:

Abstract & general comments

  • Some general rephrasing and grammar corrections are needed e.g., line 15-16 is not a clear sentence.

We edited the sentence.

  • Please reword the sentence starting Line 16 – I would disagree that pastoralism is unpopular. It is a common method of farming in many parts of the world.

We edited the sentence since it was written too simplistic. We believe it is not an easy job because it requires very special skills to be alone and continuously visually monitor animal behavior.

  • Please review all references and ensure they fit the journal’s requirements. E.g., An author’s first name does not need to be included (Line 110, 170 etc)

We have removed it for the revised version.

  • There is an inconsistency with the use of comma and period decimal points throughout. Please amend

We fixed it.

Introduction

  • Giovanetti et al [15] was done on sheep (Line 100). It currently reads like this research was done on cattle. Please clarify. You might also like to include a short discussion about the difference in detection between the two species (e.g., inertial differences due to the size of cattle over sheep) – seeBarwick, J., Lamb, D., Dobos, R., Welch, M., & Trotter, M. (2018). Categorising sheep activity using a tri-axial accelerometer. Computers and Electronics in Agriculture, 145, 289-297.

We did it.

  • Please specify which country the Douro region is in

We did it.

  • You may wish to include further discussion around the welfare benefits of sensor use (Line 146-152) – see Fogarty, E. S., Swain, D. L., Cronin, G. M., & Trotter, M. (2019). A systematic review of the potential uses of on-animal sensors to monitor the welfare of sheep evaluated using the Five Domains Model as a framework. Animal Welfare, 28, 407-420. https://doi.org/10.7120/09627286.28.4.407

We decided to not extend to much animal sensorization subsection, since it is not the focus of our work.

  • Please include further discussion as to why you have chosen to test the model on a lamb. Using the literature, explain what differences there are and why the model may perform differently on a younger animal. You haven’t yet provided sufficient background information as to why this particular research question has been included.

There was an extra collar and an additional animal, a younger one. Our purpose was to validate the sensors we would need, and the collar in the lamb was a shot in the dark. We preferred to identify it as a future work, since it requires additional analysis of the literature and tests to be repeated with several animals (adults and youngsters).

Materials and Methods

  • The access date in Footnote 1 is incorrect (page 4)

We fixed the mistake.

  • Remove reference to ‘both females’ (line 217). It is stated above.

We did it.

  • Change 17pm to 1700 or 5pm (Line 235)

 We did it.

  • Change “spawn” to “span” (Line 238)

We did it.

  • Spelling of “angles” (Line 258)

We did it.

  • Please include information on the development of the ML algorithms here. What was your training/test split? How did you split the data? Please also include a description of precision, recall and f1score as you have done for balanced accuracy.

Regarding the splitting of the data, an explanation was written at the end of the 3.3 Pre-processing and Data Cleaning sub-section.

Regarding the description of these metrics, a sub-section (3.4 Algorithm Choice and Model Evalutation) was created to explicitly describe them.

  • There has been a number of different features used in behaviour algorithm development in the literature. More details are required on the features used, parituclarly the dynamic acceleration of each axis (Line 319). Consider including an equation and/or a reference to another paper that provides this information.

We did it on section 3.3 Pre-processing and data cleaning (ref [7]).

Results

  • Figure 2 (and any others) should have the two figures referenced as (a) and (b) with a caption description that covers both

 We changed it.

  • Figure 4 is difficult to read and interpret. Consider splitting out the graphs into a faceted graph or changing the colour scheme. I also cannot see the line for All features.

We edited the color scheme to improve the understandability.

  • Table 6 is confusing to read. For example, in the row titled “No gyroscope”, the sensor cost is then listed as $7, which we know is the cost of the gyroscope from your introduction. This leads to confusion. Please amend the table so it is easier to comprehend. I suggest you also include the previous best model in this table for easier comparison.

 After reading the comment we agree it would improve understandability and we completely edited the table and renamed the feature sets to increase understandability. The table was edited to show sensor features' impact on learning accuracy, when compared with AllFeatures.

  • Line 358. I suggest you use a different word than “notorious”. It is not a correct description. The increase in accuracy is marginal at best.

 In that context, “notorious” was meant to be relative to all the metrics, including sensor cost. Notwithstanding, the wording has been updated, to better reflect the points we intend to make.

  • You may benefit from a different naming convention for your models and a Table that summarises them. It is difficult to determine which model is being discussed in the current paper.

 We did it, the names were changed in order to add an understandable meaning to the name.

  • The sentence from Line 375-377 either requires rewriting or further clarification as to what it means to have a high precision but low recall.

 We edited the sentence, which, in addition to the explanation regarding these metrics in a previous section, should help in better understanding the points made.

  • Table 7 & Table 8 – what does support refer to?

Support refers to the number of occurrences for each state, and we edited the text to highlight its relevance in terms of learning and testing.

Could the author please confirm that all the results presented for the development of the model are for the test data set? Not the training set?

 A clarification was written on the first paragraph of the Results section.

Discussion

  • Line 385-386. Is this suggestion of the heat transfer when lying the author’s hypothesis. Or is there supporting data for this? Further discussion is needed.

 This is an author hypothesis, since there is not enough data to confirm it, and we pointed it out as a future work. The text was changed accordingly. 

  • Please include further discussion as to why tree complexity is or is not beneficial in terms of algorithm development and use (Lines 387-397). You have not explored this fully and your paper would benefit from further discussion.

 We did it.

  • Please also provide further discussion as to the limitations of this work with regard to the small sample size (number of animals & data collected). It is briefly stated from Line 420, but further discussion of these limitations is necessary.

It was done.

  • Please provide further discussion as to why the author’s believe the algorithm did not work on the lamb. You have briefly discussed this from Line 424 but further discussion would be beneficial.

It was done.

Conclusion

  • Suggest you specify ‘autonomous’ monitoring or the use of sensors. Not just ‘monitoring’. Monitoring as a general concept can be performed by visual observation.

Yes, we did it.

Round 2

Reviewer 1 Report

Good improvement, most of the recommendations were attended.